# Nurses’ Experiences of Care in Portuguese Nursing Homes during the COVID-19 Pandemic: A Focus Group Study

**DOI:** 10.3390/ijerph20166563

**Published:** 2023-08-11

**Authors:** Helga Rafael Henriques, Tiago Nascimento, Andreia Costa

**Affiliations:** 1Department of Fundamentals of Nursing, Escola Superior de Enfermagem de Lisboa, Nursing Research, Innovation and Development Centre of Lisbon (CIDNUR), 1600-190 Lisbon, Portugal; 2Nursing Administration Department, Escola Superior de Enfermagem de Lisboa, Nursing Research, Innovation and Development Centre of Lisbon (CIDNUR), 1600-190 Lisbon, Portugal; 3Department of Community Health Nursing, Escola Superior de Enfermagem de Lisboa, Nursing Research, Innovation and Development Centre of Lisbon (CIDNUR), 1600-190 Lisbon, Portugal

**Keywords:** COVID-19, nursing, nursing homes, infection control, decision-making

## Abstract

The COVID-19 pandemic has had a considerable influence on long-term care facilities, exposing the shortcomings of nursing homes in implementing recommendations by health authorities. It also emphasizes the need for a nursing management model customized to the vulnerable status of residents, organizational demands, and occupational nursing requirements. We aimed to characterize the perspectives of nurses who have experienced COVID-19 in nursing homes regarding measures implemented to avoid or manage outbreaks in that environment. An interview was conducted with a focus group following the consolidated Criteria for Reporting Qualitative research guidelines. Data analysis was performed using WebQDA software following a thematic category orientation. Eight Portuguese nurses working in nursing homes from the country’s central area participated in this study. We identified three major significant areas for long-term care respiratory outbreak management: strategic (policy, staffing, and resources); tactical (training, organization, engagement, and supervision); and operational planning (vigilance, prevention of disease spread, and family involvement). From the participants’ view, the management of COVID-19 in nursing homes must be highly supportive and responsive, offering resources to control risks, supporting residents’ care, and ensuring the safety and well-being of residents and staff members. Saturation was not reached; thus, further research is needed in this area.

## 1. Introduction

The COVID-19 pandemic has significantly impacted long-term care facilities such as nursing homes and assisted-living communities. Since the first cases of COVID-19 were reported in Wuhan, China, at the end of 2019, the world has witnessed a pandemic that has affected more than 670 million people and caused approximately seven million deaths [1]. The overall notification rate of COVID-19 cases in nursing homes in the European Union/European Economic Area remains high (632.9 per 100,000 inhabitants, 17% of the pandemic maximum), although there is a trend toward decreasing incidences and hospital admissions [2]. Nursing homes are susceptible to respiratory outbreaks, such as COVID-19, because residents are mostly older adults who, due to their health conditions and advanced age, are a more vulnerable group with an increased risk of hospitalization and death [3,4]. Architectural features, high occupancy rates, shared living spaces, and people with cognitive and behavioral disorders also hinder social distancing, leading to high attack rates among residents and employees [5,6,7]. Despite extensive COVID-19 vaccination programs in many countries targeting nursing home residents and staff, there remains a need for preventive and control measures to protect the most vulnerable people from COVID-19 [8].

The COVID-19 pandemic has presented many challenges to nurses working in nursing homes. Nurses are at a higher risk of contracting COVID-19 due to the nature of their work. They are in close contact with infected patients and may not have access to adequate personal protective equipment (PPE) to protect themselves [9]. The pandemic led to high staff absenteeism due to illness or quarantine, which strained the facilities’ ability to provide adequate care to residents. It put additional pressure on the remaining staff, including the nurses [10]. Caring for critically ill residents with COVID-19 impaired their mental and emotional well-being [11,12]. Nurses were required to take on additional responsibilities without adequate training, such as administering COVID-19 tests, contact tracing, and isolation protocols, which led to feelings of insecurity. It was overwhelming and led to post-traumatic stress symptoms and burnout [12,13].

Residential care represents a unique reality within the context of healthcare services in Portugal, responding to the needs of aging people who cannot stay home but do not need hospitalization. They cover a range of acute, chronic, palliative, and rehabilitative healthcare services. According to Ordinance No. 67/2012 of March 21, in Portugal, nursing homes are organizations for collective accommodation, for temporary or permanent use, in which social support activities and nursing care are carried out [14]. This equipment ensures 24-h care. Legal rulings stipulate that nursing homes’ personnel, including nurses and nursing assistants, must conform to long-term care institutions’ capabilities, characteristics, and operations [14]. In these contexts, nurses are only sometimes on duty for a few hours regarding the 24-h possibility due to the lack of legal obligation. It implies that, especially during the night shift, nursing assistants provide care exclusively. During the COVID-19 pandemic, the shortage of human resources registered in Portuguese nursing homes before 2020 worsened.

The COVID-19 pandemic revealed nursing homes’ shortcomings in carrying out health authorities’ recommendations for assisting aging people and highlighted the need for a nursing management model tailored to the vulnerable state of residents, organizational needs, and occupational nursing requirements. Decision-making on healthcare management in nursing homes regarding workflow is dependent on the external environment of the organization (legislation, available technology, and social factors); involves the establishment of objectives; and has an implicit decision, in advance, on what to do, when to do it, and whom to involve. These authors asserted a hierarchical relationship between the three organizational levels regarding resource planning and control. When higher-level judgments place restrictions on lower-level judgments, top-down interaction is evident. By contrast, bottom-up engagement is demonstrated by feedback on the effectiveness of healthcare delivery, which aids in making higher-level decisions [15].

Therefore, decision-making in managing the pandemic in nursing homes was integrated with public health concerns adjusted to the context, including preparing human resources, making the necessary material resources available, and summarizing key messages to communicate with professionals and the public [16]. However, unlike the health sector, where clinical trials quickly began, in nursing homes (a mainly social sector), decision-makers needed more scientific evidence on the measures to respond to the pandemic [17].

Hulshof et al. [15] offered a straightforward framework for healthcare managers and researchers to identify, analyze, and categorize planning and control decisions. However, in the operational, tactical, and strategic organizational dimensions, little is known about the planning and control of resources in nursing homes, particularly the response to the needs created during the pandemic [15,18]. Developing an intervention model to prevent and manage pandemic outbreaks in these organizations is essential.

Nurses play an essential role in the fight against COVID-19. They have been on the front lines, helping to care for victims and stopping the spread of the infection. Therefore, it is crucial to reflect on what we can learn from their experiences, paying particular attention to people, the resources of the organization, and the sustainability of the alternatives developed [19]. This study aims to characterize the perspectives of nurses who have experienced COVID-19 in nursing homes regarding the measures that must be implemented to avoid or manage outbreaks in that environment. Since the scientific evidence is sparse in this area, we intend to provide an initial exploration of the topic and generate preliminary insights.

## 2. Materials and Methods

This study is part of the Educovid Project, a mobile application for an integrated response to COVID-19 in Portuguese residential structures for aging people. In phase 1, we started by reviewing the literature to systematize the evidence about the infection control measures to be adopted by nursing home staff to minimize the risk of transmission of COVID-19. This is a phase 2 study, in which a qualitative exploratory approach was followed to gain an in-depth understanding of participants’ perceptions of the interventions, policies, and measures that nursing homes should implement at the strategic, tactical, and operational levels to prevent or manage COVID-19 outbreaks in nursing homes. We intend to see whether it aligns with the above systematic review and, in phase 3, reaches a consensus with a Delphi panel that will involve nurses from all over the country.

Qualitative studies study phenomena within their natural settings, aiming to understand and interpret them based on the meanings attributed by the individuals involved [20] A single interview was conducted with a focus group. It is a specific qualitative research design that involves bringing together individuals from the study population in a designated setting to engage in a group discussion centered around a particular topic or issue [21,22]. This research approach aims to generate in-depth data by tapping into the participants’ collective perspectives, experiences, and insights [21,22]. We follow the consolidated criteria for reporting qualitative research (COREQ) guidelines [23]. In particular, we provide information about the composition of the research team and their backgrounds; we describe the overall study design, including the rationale for using a qualitative approach and the methods used to collect data; we detail the criteria used for participant selection and the recruitment process; we provide a clear description of the data collection methods; we explain the procedures used to analyze the qualitative data; and we address the strategies used to enhance trustworthiness, such as member checking.

Sampling/Participants. This study was conducted with senior nurses working in Portuguese nursing homes during the COVID-19 pandemic in the country’s central region. Study participants were recruited for convenience, from among the contacts of some researchers, following the snowball effect [24]. This initial contact was made by telephone, asking for informed consent to participate in the study. It was expected that there would be six to ten participants [25]. Considering the potential cancellations, we recruited 11 nurses.

Data collection. A 10-item questionnaire was applied one week before the interview date to assess the sociodemographic and professional characteristics of the participants: age, gender, educational and professional qualifications, years of professional experience, years of experience working in long-term care, exclusivity in nursing homes, and weekly hours of work in nursing homes. Regarding the nursing homes where participants worked last year, they were asked about 24-h face-to-face nursing coverage and which professional groups ensured the nursing homes’ governance. This questionnaire was developed through a review of the existing literature, establishing widely recognized categories and response options and leveraging the expertise of the researchers in the field. The questionnaire was emailed to all potential participants, and informed consent was obtained. Only the participants who submitted the questionnaire were called for interviews. The starting point was known to the participants during this phase.

Given the COVID-19 pandemic’s restrictions on circulation, a synchronous online focus group was planned [22]. The online session was recorded with the participants’ prior consent. One PhD researcher (female) with previous experience in conducting focus groups operated as the moderator (AC), while two other researchers worked as rapporteurs (HRH and TN), examining the focus group’s conduct and taking field notes that would help contextualize the focus group for later data transcription. The moderator had no prior relationship with the participants and introduced herself as a researcher with a special interest in the field of the aging population.

The focus group began with a welcome from the moderator, who also expressed gratitude for everyone’s availability before explaining the session’s goal and asking permission to record. The moderator ensured that all data were anonymized and that no personal information was included in the interview transcription. It was reinforced that all the materials would be destroyed five years after the end of the study. The rules inherent to the session were explained, and the moderator asked each participant to introduce themselves. After this moment, an opening question was asked: “*I will give you a few minutes to think about your experience of providing care in nursing homes during the COVID-19 pandemic. I would like to hear from you about measures that you consider essential to implement in the nursing homes, where you operate. Would you like to share your experience?*” In addition, a set of guide questions was developed to facilitate the discussion, stimulate the free sharing of opinions, and allow the emergence of new topics to be explored by the moderator.

Data analysis. Audio transcription was performed using the software Sonix ^®^ AI version 2021. Data analysis was performed on the WebQDA platform (version 2017) following a thematic–categorical orientation according to the content analysis methodology [26]. The themes and categories were divided into three major areas: strategic, tactical, and operational planning, as suggested by Hulshof et al. [15] for decision-making in healthcare settings. These themes provided an initial framework to guide our exploration of the data. It is important to underscore that our analysis approach did not enforce predetermined assumptions on the findings. Instead, we allowed the subthemes to naturally arise from the data, fostering an open and unbiased interpretation. This approach facilitated a comprehensive understanding of the participants’ perspectives and ensured that the findings accurately reflected their experiences [20]. Combining deductive analysis guided by predetermined themes with inductive analysis driven by the data, we aimed to balance the existing theoretical frameworks and the unique insights derived from our participants’ responses. This approach allowed us to capture anticipated and unanticipated themes, enriching our findings. Each participant (P) was coded with a number to protect their identity.

Two investigators (TN and HRH) independently coded the transcripts and then compared the codes. Revisions to the codebook were made using selective coding, and the codes were consolidated into categories. An additional investigator (AC) assisted in reviewing and forming a consensus between the categories and emerging themes.

Ethical considerations. The Ethics Committee of the Nursing School of Lisbon approved this study. Participants received guarantees of confidentiality and anonymity and the freedom to leave the study.

## 3. Results

Eleven nurses were approached to participate. Nine signed consent forms, and eight completed the interviews. One person did not attend because of personal impediments. The focus group lasted 120 min and took place in December 2020. The sample consisted of eight participants (five females and three males). The participants had a mean age of 45.9 years (SD = 7.1) and an average of 22.5 years of professional experience (SD = 6.3), of which an average of 10.5 years (SD = 9.4) were in nursing homes. Only two participants worked exclusively in nursing homes, and the others worked an average of 20.3 h per week as a second job. Only one participant had a master’s degree. Only one nurse worked in a nursing home with a 24-h nursing presence. Six participants work for nonprofit organizations. The nursing homes included in this study had a range of bed capacities varying from 14 to 128 beds (x̄ = 78.25; SD = 48.5) and were in the central region of Portugal (Table 1).

A qualitative analysis of the transcripts identified 141 enumeration units (codes), which resulted in three themes, ten categories, and ten subcategories. The transcripts were not returned to the participants. The major categories were incorporated into a theoretical framework to guide future research. The key themes are described below and illustrated using anonymized quotes (Table 2).

### 3.1. Theme: Strategic Planning

Strategic planning integrates categories that concern planning related to organizational management structures. The categories of management policy, endowments, and the management of physical and technological resources are part of this domain.

#### 3.1.1. Management Policy

The participants highlighted that nursing homes typically have policies that outline how facilities are managed and operated. These policies cover many topics, including resident care; medication management; staffing; safety and security; finances; and compliance with national and international laws, regulations, and policies. Risk management in nursing homes during the COVID-19 pandemic must be evaluated to ensure the safety and well-being of residents and staff. In Portugal, social care professionals primarily hold technical direction roles in nursing homes. However, some informal movements advocate for nurses to play a more significant role due to increasing healthcare demands and complex aging population needs. These movements argue that nurses’ specialized knowledge, including clinical assessment and care planning, can enhance residents’ well-being and safety. Integrating nursing expertise in a technical direction can improve the overall care quality and healthcare outcomes. A multidisciplinary perspective can provide a 360-degree view, considering all angles of the problem:

*“It is now widely advocated that technical direction of nursing homes should be limited to nurses”. I do not think so. I think the model of technical direction should be different because technical direction does not have to be based on just one person, that is, “what is social is for social areas, what is health is for health areas, and so on… all can contribute and learn from each other.”* P7

This acknowledgment aligns with the understanding that successful management policies should not be limited to one specific area but should consider the interplay and integration of various disciplines and perspectives. It emphasizes the need for collaboration, cooperation, and shared decision-making between social and nursing professionals, ultimately leading to more holistic and effective management practices.

#### 3.1.2. Staffing Ratios

The COVID-19 pandemic has significantly impacted staffing ratios in long-term care settings owing to illness and quarantine requirements. These low ratios/unsafe staffing create an insufficient response to nursing care. It highlights the need to discuss the issue of delegating nursing tasks in nursing homes:

*“… it is important to see the ratios regarding the number of female nursing assistants to provide the proper care”* P3

*“The problem only arises because we do not have full-time nurses… so we have to delegate all those that are not invasive and highly complex….”* P7

In Portugal, delegating nursing functions involves assigning specific tasks and responsibilities to nursing assistants. In the healthcare field, nursing assistants work under the supervision of nurses and provide essential care and support. However, in the social field, the delegation of tasks can vary, as social workers or nurses may supervise nursing assistants. During the COVID-19 pandemic, nurses took a more active role in supervising assistants as healthcare needs increased. There have been notable changes in the delegation of functions in nursing homes. Nursing assistants have implemented screening measures, such as temperature checks and symptom monitoring, for both residents and staff. They have received additional training and support to implement strict infection control protocols, including proper hand hygiene, personal protective equipment (PPE), and enhanced cleaning and disinfection practices.

Interdisciplinary collaboration has also been prioritized during the pandemic. Nursing assistants have worked closely with nurses, doctors, social workers, and other healthcare professionals to ensure coordinated care and prompt responses to the needs of residents.

These adaptations in the delegation of functions reflect the unique challenges posed by the COVID-19 pandemic and highlight the importance of teamwork and collaborative efforts in providing safe and effective care in nursing home settings.

#### 3.1.3. Management of Physical and Technological Resources

Two subcategories stand out in this category: the organization of spaces and technology. Nursing homes were forced to reorganize their care spaces, creating areas and circuits to circulate people and materials or isolation areas for infected residents to minimize the transmission of the virus (cohorts of positive cases):

*“We immediately decided to close one of the wards and concentrate all infected users”* P1

*“They [residents] stayed in the basement, and the visitors stayed on the ground floor balcony and could see each other.”* P6

The information system is viewed as a facilitator of information management among nurses. The pandemic called attention to this need for those unfamiliar with information systems.

*“(…) and having a computer application where they [staff members] look for a certain symptom leads, for example, to put a person in isolation. It will avoid the call at four in the morning (…). Current information systems allow recording information, allowing them to access information, but they do not tell them how to act in situations and with that possibility. I think that was something interesting.”* P7

### 3.2. Theme: Tactical Planning

In the tactical planning theme, the categories reflect the strategy used in nursing homes regarding human resource management: team training, team organization, team engagement, and supervision.

#### 3.2.1. Team Training

The team training category included the content, strategies, and trainees’ specificities subcategories. It is essential to provide proper training on infection control measures:

*“(…) we did train in Infection Control as well as hand hygiene care, everything that is due to that point (…) where we passed on and reinforced all the needs in terms of the placement of PPE’s and care to be taken in infection control.”* P3

*“(…) the disease, how could we prevent it, the respiratory etiquette (…). When did they obligatorily have to wash their hands or sanitize with SABA [alcoholic-based aseptic solution]. (…) putting on and removing PPE with those residents (…).”* P2

*“(…) our nursing assistants and the entire team were trained to wear, remove, circulate, and not to have direct contact with other users who were not assigned to them.”* P1

The participants understood, however, that several other areas of care need attention, not only when responding to COVID but in all circumstances. Among these contents are topics related to communication, the humanization of care, first aid and basic life support, falls, and wound care. The educational approach employed encompassed procedure demonstrations, simulations, and training. The participants opted for informal contexts for training using peers to achieve all the nursing assistants:

*“I identified a nursing assistant, whom I ask to train the others [nursing assistants], and daily support them”* P3

The trainees’ specificities determined the strategies adopted and the depth of the content. The trainees (nursing assistants) had significant qualification deficits:

*“(…) our nursing assistants are always from low education and higher age groups.”* P3

#### 3.2.2. Organization of Teams

Participants in the organization of teams category reported strategies for addressing human resource needs. There was a need to create fixed teams involving professionals during all shifts and to build mirrored scales: “We were working on a mirror scale, (…) with shifted schedules…” P1

*During the early stages of the pandemic, complete retention of employees at nursing homes was one of the strategies used to ensure care. Nursing teams felt enormous pressure to respond to different requests, given that the nursing care needs to be increased significantly: “Our usual functions were neglected; that is, they were uncovered. We managed to replace everyone, we managed to replace everyone within all functions within nursing homes, but no one could replace us.”* P4

#### 3.2.3. Team Engagement

Participants discussed how to motivate nursing assistants and how they became involved in the organization’s mission:

*“I tried to make people realize how important they [nursing assistants] were.”* P5

#### 3.2.4. Supervision

Participants highlighted the role of nurses in direct or indirectly monitoring nursing assistants’ care:

*“They need supervision on the assessment of vital signs and the medication administration”* P1

*“Because a nurse cannot assist the five tube-feeding residents while also assisting the numerous diabetics, we have to delegate… we have to read all of these, all competencies, and effectively distribute these tasks.”* P3

### 3.3. Theme: Operational Planning

Operational planning for nursing homes during the COVID-19 pandemic involves implementing strategies to protect residents and staff from infection while maintaining a continuity of care. The operational planning theme included categories related to care management: resident vigilance, prevention of the spread of the disease, and family involvement.

#### 3.3.1. Residents’ Vigilance

The resident vigilance category included in-person and remote subcategories. Face-to-face vigilance refers to nurses developing in-person actions for systematic assessment, symptom monitoring, and problem identification. This form of intervention was highly valued by the participants, as it not only ensured the necessary care but also contributed to better risk management:

*“Our nursing assistants are very dependent on nursing [related to resident vigilance] because as we are there every day from Monday to Sunday from 9 am to 9 pm, the assistants who work in the morning feel safe (…).”* P3

Remote vigilance concerns the moment when an institution does not have the physical presence of a nurse. During this period, nurses delegated several caring tasks.

*“(…) medication is prepared exclusively by the institution’s nurses, but then we delegate the administration to nursing assistants. Why? Because they are the ones who are there to give medication for fasting or supper. I mean, impossible not to delegate.”* P7

*“(…) it is important to give skills to the nursing assistants because not all homes can have 24-h nursing.”* P2

#### 3.3.2. Preventing the Spread of Disease

This category involves the subcategories of residents (re)admissions, index case identification, and PPE. The admission process is essential for teams, residents, and family members. It became a source of fear due to suspicion or a case confirmed a posteriori:

*“(…) we started to do prophylactic isolation to all users who went to the emergency service, even if it was six hours because they could get infected there.”* P2

*“We have a situation that worries us: a dialysis user who goes to the hospital three times a week. This situation worries us because there have been cases of COVID-19 in hemodialysis. Fortunately, he has been doing well thus far and is also doing all the isolation (…).”* P6

During an outbreak, index case identification was a concern for identifying the transmission chains (suspected and confirmed cases). Nursing homes built contingency plans for confirmed or suspected cases in which preventive and corrective actions were defined.

Preventing the spread of the disease requires significant investment in personal protective equipment. During the pandemic, PPE use has become a complement to uniformity. Management and infection control associated with the placement and removal of this equipment has become the focus of professional concern, with a more significant expenditure of time and a need for constant verification to reduce or eliminate exposure to SARS-CoV-2:

*“So there is also enough material besides being a private home. There is enough PPE material for them to use.”* P6

#### 3.3.3. Family Involvement

The family involvement category is related to new ways for family–resident interactions. Family visits to users were temporarily suspended to reduce the risk of the virus entering nursing homes through families:

*“We use a tablet to make a call, sometimes with our assistance, sometimes with the assistance of those who know how to make and carry it.”* P5

*“At this time, it is very complicated in terms of saying goodbye to family members, although we have the possibility of family members visiting (…) it is through the acrylic glass, and they can see and talk to each other, but there is no contact between them”* P2

The family involvement category highlights the implementation of new methods for family–resident interactions in response to the temporary suspension of in-person visits due to the risk of virus transmission.

## 4. Discussion

According to participants’ perceptions, the management of nursing homes during the COVID-19 pandemic involved strategic, tactical, and operational planning, which provided new insights and contributed to the existing knowledge in the field [15].

Our results suggest that a multidisciplinary team, including nurses, should ensure strategic management. This perspective is consistent with other studies [27,28] that emphasized the importance of an interdisciplinary team in developing an integrated care approach. With integrated care, healthcare professionals from different disciplines share information, collaborate, communicate with team members, and coordinate comprehensive care for residents [27,28]. The complexity of the care situations experienced in nursing homes requires an integrated view of different disciplines and coordinated care between public health departments, long-term services, support agencies, and nursing assistants [29].

Nursing homes have adopted a reactive management policy. A reactive management policy involves responding to events or situations as they occur, rather than taking proactive or preventive measures. It entails addressing issues as they arise, often ad hoc or unplanned, instead of having preestablished strategies or preventive actions in place. In Portugal, like other regions, this response is conditioned by government decisions and public health measures. One of the decisions substantially impacting nursing homes is the limitation of professionals/nursing assistants to a single work institution during the pandemic. Since many professionals work in nursing homes as a second job, it has moved many staff away from these facilities [30]. Thus, at the top of the concerns regarding management policy is the safe allocation of human resources to relocate comorbidity situations due to COVID-19 [31,32,33]. In Portugal, as determined by the government to meet these needs, whenever nursing homes were unable to respond to these needs, they could activate Rapid Intervention Brigades, composed of doctors, nurses, nursing assistants, and service assistants recruited and managed by The Red Cross in concert with Social Security [34]. Despite the creation of this response, human resources for these brigades were also scarce, considering the need during active outbreaks in nursing homes.

The pandemic has heightened the healthcare needs of nursing home residents, transforming many of these facilities into acute and post-acute care facilities. Despite emerging health needs, not all nursing homes are staffed with health professionals, forcing institutions to turn their policies towards the immediate training of formal nursing assistants and create strategies to improve the delegation process and supervision of these nursing assistants. In this context, nurses played a decisive role not only in the vigilance and monitoring of residents but also in the training and supervision of nursing assistants in particular aspects of care, such as measuring body temperature, oxygen saturation, respiratory rate, tension blood pressure, heart rate, and health status change [35]. The delegation of functions in nursing homes is a central component in care delivery; therefore [36], in a pandemic situation, it made sense to create a plan for delegating tasks to those responsible for communicating with team members or family [37].

Although incipient, information systems are essential for monitoring outbreaks and accelerating the supply of PPE, additional employees, and medicines in nursing homes [38]. As other studies concluded, the COVID-19 pandemic has dramatically accelerated research on integrating digital technologies and healthcare [39].

The tactical dimension of management in nursing homes fits with the complete management of teams. These categories fit into the definition of tactical planning, which can be defined as the extension of operational management with a higher degree of supervision [40]. From a functional standpoint, tactical planning allows for real-time response capacity adjustment considering the variability in contexts and available resources [41]. Tactical planning increases the organization’s structural resilience and facilitates the restructuring and reorganizing of established practices. It introduces an innovative approach that may require implementation time and may only sometimes result in immediate changes [42].

In an aggregated way, developing team members’ skills and involvement in the training process reduces the need for a decision-making support structure [43]. In the context of a pandemic, it has become central to training concerning infection control [37], symptom vigilance [44], and improving strategies for communication and the humanization of care [45,46]

In the context of a pandemic, training activities are critical for the rapid development of essential skills. New technologies can play a singular role in allowing information systematization and access [47].

Our findings showed that the organization of teams was consistent with other studies, such as innovative scheduling, shifting mealtimes, and increasing periods of permanence in nursing homes to reduce personnel entry and exit, increase containment, and implement a shifting mirror system [48]. In this scheduling arrangement, two or more teams work opposite shifts, with one team working during specific hours and the other working during complementary hours. The goal is to ensure continuous coverage and minimize the risk of cross-contamination or transmission by reducing direct contact between different teams. Hiring new staff to respond to the lack of human resources, especially nurses, also conditions the organization of teams [49,50].

Health authorities usually lack coordination in communicating the vast amount of new guidance related to the response to COVID-19 in nursing homes. Our results showed that nursing home leaders were essential in aggregating information and implicating the staff in decision-making. Nursing assistants’ participation in decision-making helped foster a sense of ownership and commitment to the response efforts and identified and addressed any issues that may arise [51]. This category of results was one of the most surprising in the context of the pandemic and avoided several constraints on the scarcity of human resources.

According to our research, nurses are crucial for supervising the care and management of nursing homes. Nursing assistants appreciated the nurses’ contributions to risk assessment, strategy development, and safety enhancement through careful supervision. By raising their workload in these situations and validating tasks that could not be delegated, nurses reinforced their rules for operating in nursing homes [52,53].

Confronted with the taxonomy of Hulshof et al. [15], our results highlight new areas for tactical decision-making, such as nurse supervision and team building, involving nursing assistants.

Operational planning concerns decision-making about providing residents with safe healthcare [15]. The prevention and management of outbreaks in nursing homes involve the remote and face-to-face vigilance of residents, prevention of the spread of the disease, and family involvement. Except for the treatment schedule, which we can consider to be part of (but not exhaustive of) the category of the “vigilance of residents”, the remaining categories are not represented in the Hulshof et al. [15] taxonomy.

Residents’ vigilance concerns the participants because of the harmful complications of COVID-19 [15]. Vigilance is an essential component of nursing care that includes detecting clinically relevant signs and symptoms, analyzing them, and identifying the existing risks [51]. It presupposes assessing the need for care and determining the human resources necessary to respond to such demands [15].

Like Hulshof et al. [15], our findings focused on admission policies. However, during the pandemic, there have been some concerns about admissions because of the risk of spreading the SARS-CoV-2 infection, which does not exist in other situations. Measures adjusted to residents’ (re)admission, identification of the index case, use of equipment protection, and family involvement measures were not considered in the taxonomy of Hulshof et al. [15].

During the COVID-19 pandemic, the universal and systematic testing of residents and professionals was an essential vigilance measure to identify new cases, especially when there was high community transmission [8,52,53]. Screening based on symptom or temperature assessments and point prevalence testing was considered ineffective in preventing SARS-CoV-2 infection, given the high number of people with asymptomatic infections [8]. However, the rapid clinical deterioration of infected people has led to the vigilance of the patient’s condition as a vital control measure [54]. New technologies have played a decisive role in this field by promoting innovative communication and remote care [55,56]. Furthermore, according to the participants’ perceptions, improving risk management and real-time response in outbreak situations [8] and preventing the spread of the disease are present concerns in nursing homes. The confinement of staff and residents was a measure that reduced the mortality from COVID-19 and the number of people infected in nursing homes [8]. The measures adjusted to residents’ (re)admission, identifying the index case, and using PPE aligned with the top international recommendations [57,58,59,60,61]. As Hulshof et al. [15] defended, they are highly dependent on the decision-making of the strategic and tactical dimensions. Therefore, decisions are of low flexibility during operational planning, as they are defined at higher levels.

Family involvement in care came to be viewed as threatening to residents’ safety. The restriction or conditioning of visits (frequency, duration, and volume) is widely used in nursing homes. Hand hygiene, temperature and symptom screening, physical distancing, and the use of masks were additional measures aimed at visitors who became part of the daily life of nursing homes [62,63]. Restricting or conditioning visits lead to social isolation, which negatively impacts residents’ well-being [3,64]. The pandemic is also associated with high anxiety, depression, post-traumatic stress, sleep disturbances, and functional decline [64].

During the pandemic, operational planning required identifying vigilance care requirements, preventing disease spread, and family involvement. This planning included, in addition to the perspective of the resident/family, the dimension of the organization of care (professional working hours and available resources) in a coordinated work logic and continuity with other health organizations (for reporting cases, monitoring, the referral of patients, and (re)admission of residents).

The strengths of our study included its unique findings and the use of rigorous qualitative research methods that allowed the expansion of the framework proposed by Hulshof et al. [15]. Although the focus group methodology is defined by the proximity of the researchers and the participants, this study was conducted online, which meant that we faced limitations that influenced the fluidity of the discussion, which differed from face-to-face discussions. These constraints might have inhibited participation. The participants were not representative of all nurses working in nursing homes in Portugal. Thus, the results of this study were valid only for the group of interviewed participants. The sample size was small, which is common in qualitative studies. Although the participants were recruited from various nursing homes, the results did not represent Portuguese reality, which might also have limited the generalizability.

## 5. Conclusions

The COVID-19 pandemic has exposed long-term care facilities to various shortcomings. Understanding the viewpoints of nurses who have dealt with the pandemic’s early stages and the steps that must be taken to prevent or control outbreaks may help leaders make better decisions in the future. Using a focus group methodology made it possible to identify three significant dimensions of nursing home management during the pandemic: strategic, tactical, and operational planning.

From the participants’ view, the management of COVID-19 in nursing homes must be highly supportive and responsive, offering resources to control the risks, supporting residents’ care, and ensuring the safety and well-being of residents and staff members. Participants suggested that a multidisciplinary strategy (including nurses) for decision-making at the organizational level is better suited to the needs of professionals and residents. Nurses also play a critical role in supervising nursing assistants and team building. Nurses are better positioned to understand patients’ and families’ needs and to find the best way to involve each one in their care plan.

To the best of our knowledge, no research supports decision-making in nursing home management during pandemic outbreaks from the nurses’ perspective. Although the participants’ perceptions reflect the local Portuguese reality, the analysis of the results and their partial confrontation with the available international evidence shed light on this issue. These findings led to the development of a solid and scientifically supported model for infection prevention and control in nursing homes during outbreaks. We believe that saturation was not reached; thus, further research is needed in this area. However, our research aimed not to make sweeping generalizations or claims based solely on the findings of one focus group. Instead, the focus group was an exploratory tool to gather rich qualitative data and generate initial insights into the research topic. Our results serve as a starting point for further research. Combined with a literature review, the insights gained from the focus group can inform the development of a Delphi panel and the generation of an expert consensus, the next step of our project. In phase 3 of the Educovid Project, we are working on a national Delphi panel to reach a consensus on this topic.

## Figures and Tables

**Table 1 ijerph-20-06563-t001:** Participants’ characteristics.

Characteristics		N
Age (years)	Mean	45.9
	SD	7.1
Gender	Female	5
	Male	3
Educational qualifications	Graduate nurse	7
	Master degree	1
Professional experience (years)	Mean	22.5
	SD	6.3
Experience working in LTCF (years)	Mean	10.5
	SD	9.4
Employment status in nursing homes	Full-time (35 h/week)	2
	Part-time (mean 20.3 h/week)	6
Nursing homes with 24-h nursing coverage	Yes	1
	No	7
For-profit organizations	Yes	2
	No	6
Bed capacities	Maximum	128
	Minimum	14
	Mean	78.25
	SD	48.5

**Table 2 ijerph-20-06563-t002:** Thematic framework for nursing management outbreaks in nursing homes.

Themes	Categories	Codes	Subcategories	Representative Quotes
Strategic planning (Organizational structure level)	Management policy	9		“(…) a technical direction should not be just social or nursing, but both” P2
	Staffing Ratios	15		“COVID-19 pandemic clarified the need for nurses in nursing homes, who play a fundamental role in multidisciplinary teams.” P1
	Management of physical and technological resources	20	Organization of spaces	“We immediately decided to close one of the wards and concentrate all infected users” P1
			Information system	“(…) the nursing assistants can register and we at home, in real-time, we can understand the status of residents” P2
Tactical planning (Team level)	Team training	33	Contents	“The topic of first aid and basic life support was the most requested between nursing assistants” P5
			Strategies	“I think we should train through demonstration and experimentation, especially in practical content” P3
			Trainees’ specificities	“These people come from [their last job was in…] the countryside, they come from the aviary, they come from the bakery, they do not even come from the commerce that could be slightly more differentiated… and our main objective is to train these people so that we can effectively delegate safely.” P7
	Organization of teams	7		“We have shifted schedules to go to the bathroom, go out…” P1
	Team engagement	6		“(…) one of the strategies that we used [to promote nursing assistants involvement] was to share data on pressure ulcers, falls, and referrals to the hospital” P2
	Supervision	17		“…preparation and supervision were fundamental to the early detection of possible new cases.” P1
Operational planning (Care management level)	Residents’ vigilance	17	In-person	“Our nursing assistants are very dependent on nursing [related to resident vigilance] because as we are there every day from Monday to Sunday from 9 am to 9 pm, the assistants who work in the morning feel safe (…).” P3
			Remote	“At night, (…) we are not there from 9 pm and, therefore, it ends up being a little bit on the conscience and common sense of the group of nursing assistants there.” P4
	Preventing the spread of disease	12	(re)Admissions of residents	“(…) we started to do prophylactic isolation to all users who went to the emergency service, even if it was six hours because they could get infected there.” P2
			Identification of the Index Case	“(…) some suspicions arose and in all of them we managed to perform the swab later and it was negative.” P3
			Personal Protective Equipment	“So there is also enough material in addition to being a private home. There is enough PPE material for them to use.” P6
	Family involvement	5		“We use a tablet to make a call” P7
	Total	141		

## Data Availability

The data presented in this study are available on request from the corresponding author. The data are not publicly available due to the participants’ privacy.

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
