# Peer review of "Nurses’ Experiences of Care in Portuguese Nursing Homes during the COVID-19 Pandemic: A Focus Group Study"

_ijerph, 2023, doi:10.3390/ijerph20166563_

Round 1

Reviewer 1 Report (Previous Reviewer 1)

The manuscript has improved significantly. 

Reviewer 2 Report (Previous Reviewer 2)

Dear authors,

I appreciate the effort that you have put into revising your work in line with the previously suggested comments and recommendations.

I feel that manuscript can be accepted in present form.

Best regards!

Reviewer 3 Report (Previous Reviewer 3)

Dear Authors,

I read your manuscript with interest.

The period of the covid-19 pandemic was a great test for medical workers and health care institutions. Your research shows the difficulties nursing home medical staff faced during the pandemic. These are valuable results that are worth getting acquainted with in order to further improve the quality of care, even beyond the pandemic.

I like the way you described your study. It's clear and readable to me. Congratulations on your research.

This manuscript is a resubmission of an earlier submission. The following is a list of the peer review reports and author responses from that submission.

Round 1

Reviewer 1 Report

Dear Authors.

I do appreciate your efforts in conducting the study and writing this manuscript.

Frist of all, I think the title is an important element for any study to attract readers and to show what the study is really about. It seems that you need to rewrite the title in accordance with the actual content of your study and manuscript.

Maybe it is better to think about something like the “nurses’ experiences of providing care in nursing homes during the COVID-19 pandemic” – OR – their perspectives of the nursing home care during the COVID-19 pandemic.

Wish you all the best.

Author Response

We would like to express our gratitude to Reviewer 1 for their insightful comments and suggestions. We have carefully considered their feedback and have made the following revisions in response:

R1: "It seems that you need to rewrite the title in accordance with the actual content of your study and manuscript."

Our Response: Thank you for your suggestion regarding the title. We have carefully considered your input and have made the necessary changes accordingly. The revised title now reflects your recommended modification, aligning it more closely with the aim and content of the manuscript. We greatly appreciate your valuable input, which has contributed to enhancing the overall presentation of our work.

Reviewer 2 Report

Dear authors!

Thank you for the opportunity of reviewing this manuscript. I read the article entitled “COVID-19 pandemic clarified the need for nurses in multidisciplinary teams”: a focus group with nurses from Portuguese

nursing homes." The aim of the study was to describe the perceptions of caregivers who have experienced Covid-19 in a nursing home about measures to prevent or manage outbreaks in this setting. In my opinion, the manuscript is interesting, clearly presented, and follows accepted standards. With regard to this work, the authors identify three important areas for managing respiratory disease outbreaks in long-term care: strategy, tactics, and operational planning. This study supports decision-making in nursing home management during a pandemic from a caregiver perspective. The authors correctly state all limitations. I can only suggest a small change to the meaning of the sentence on line 226, but I think the article can be published as-is. Best wishes!

Author Response

We would like to express our gratitude to Reviewer 2 for their insightful comments and suggestions. We have carefully considered their feedback and have made the following revisions in response:

R2 - "I can only suggest a small change to the meaning of the sentence on line 226"

Our Response: Thank you for your feedback. In response to your comment, we have revised the sentence in line 226 to ensure clarity and accuracy. We believe that this modification enhances the overall coherence and effectiveness of the manuscript. We sincerely appreciate your valuable input and contribution to improving the quality of our work.

Reviewer 3 Report

Dear Authors,

I read your manuscript with interest. The period of the covid-19 pandemic was a great test for all organizations, but especially for healthcare facilities.

I like the way the research is conducted, although it is known that online research has its limitations, which you mentioned.

I like that you have divided the study into 3 main thematic strands. This makes the results transparent.

Your study shows the difficulties faced by nursing homes' medical staff. These are valuable outcomes.

It seems to me that it was still possible to indicate how they should be used now, in a time without a pandemic. It would be a complement to the whole. This would be a kind of protection against possible difficulties in the future.

Author Response

We are grateful to Reviewer 3 for their valuable comments and suggestions. We have taken their feedback into serious consideration and have made the following revisions in response:

R3: "It seems to me that it was still possible to indicate how they should be used now, in a time without a pandemic. It would be a complement to the whole. This would be a kind of protection against possible difficulties in the future".

Our Response: Thank you for your valuable input. We acknowledge that incorporating this information can significantly enhance the article. As per your suggestion, we have included a new paragraph addressing the topic on lines 462-470. We believe that this addition further strengthens the argument and provides important insights into the subject matter. We appreciate your contribution and are grateful for your guidance in improving the manuscript.

Round 2

Reviewer 1 Report

comments were addressed